# AI-SearchPlanner: Modular Agentic Search via Pareto-Optimal Multi-Objective Reinforcement Learning

## Abstract

Recent studies have explored integrating Large Language Models (LLMs) with search engines to leverage both the LLMs' internal pre-trained knowledge and external information. Specially, reinforcement learning (RL) has emerged as a promising paradigm for enhancing LLM reasoning through multi-turn interactions with search engines. However, existing RL-based search agents rely on a single LLM to handle both search planning and question-answering (QA) tasks in an end-to-end manner, which limits their ability to optimize both capabilities simultaneously. In practice, sophisticated AI search systems often employ a large, frozen LLM (e.g., GPT-4, DeepSeek-R1) to ensure high-quality QA. Thus, a more effective and efficient approach is to utilize a small, trainable LLM dedicated to search planning. In this paper, we propose **AI-SearchPlanner**, a novel reinforcement learning framework designed to enhance the performance of frozen QA models by focusing on search planning. Specifically, our approach introduces three key innovations: 1) Decoupling the Architecture of the Search Planner and Generator, 2) Dual-Reward Alignment for Search Planning, and 3) Pareto Optimization of Planning Utility and Cost, to achieve the objectives. Extensive experiments on real-world datasets demonstrate that AI SearchPlanner outperforms existing RL-based search agents in both effectiveness and efficiency, while exhibiting strong generalization capabilities across diverse frozen QA models and data domains.

## 1 Introduction

Recent research has investigated combining Large Language Models (LLMs) with search engines to harness both the LLMs' internal knowledge and external information retrieval (IR), thereby enhancing performance on complex reasoning tasks. The early representative paradigm, Retrieval-Augmented Generation (RAG), retrieves relevant passages based on input queries and incorporates them into the LLM's context for generation. While this approach facilitates the utilization of external knowledge, most RAG systems rely on static retrieval mechanisms, lacking adaptive planning capabilities for complex reasoning. Consequently, their performance remains highly sensitive to retrieval accuracy. With advancements in the general capabilities of LLMs, recent research has explored leveraging search engines as tools — primarily through prompt-based methods or self-supervised fine-tuning (SFT). These approaches enable LLMs to autonomously determine whether and how to invoke search engines based on the problem-solving context. However, prompt-based methods often exhibit limited generalization, particularly for complex reasoning tasks that fall outside the LLM's pretraining distribution. Although SFT-based methods offer greater adaptability, they require large-scale, high-quality annotated trajectories of search-reasoning interactions, posing scalability challenges. Furthermore, recent studies indicate that SFT methods tend to overfit to memorized reasoning paths, further constraining their generalization to out-of-distribution scenarios.

Recently, reinforcement learning (RL) has emerged as a promising paradigm for improving the reasoning capabilities of LLMs. Models such as OpenAI-o1[1] and DeepSeek-R1 Guo et al. (2025) employ RL techniques (e.g., PPO Schulman et al. (2017), GRPO Shao et al. (2024)) to enhance logical reasoning and problem-solving through iterative experiential learning. Research like Search-R1

---

[1]https://openai.com/zh-Hans-CN/index/learning-to-reason-with-llms/

Jin et al. (2025) further highlights the potential of RL by leveraging question-answering rewards in multi-turn reasoning to train LLMs for search engine interactions. This approach yields significant performance improvements, demonstrating RL's effectiveness in enhancing LLM-based search utilization. Although RL-based search agents have demonstrated superior performance in complex reasoning tasks, existing approaches typically rely on a single LLM to handle both search planning and QA tasks in an end-to-end manner, which poses significant challenges in optimizing both capabilities simultaneously. In particular, in practical scenarios, sophisticated AI search systems such as Baidu[2] and Tencent Yuanbao[3] commonly utilize a large frozen LLM (e.g., GPT-4 Achiam et al. (2023), DeepSeek-R1 Guo et al. (2025)) to ensure high-quality QA. Thus, a more effective and efficient solution is to employ a small, trainable LLM dedicated to search planning, thereby improving QA performance while maintaining low computational latency.

To address this challenge, in this paper, we propose **AI-SearchPlanner**, a novel reinforcement learning framework designed to enhance end-to-end QA performance by focusing on search planning. Specifically, our approach introduces three innovations to achieve the objectives:

1) **Decoupling the Architecture of the Search Planner and Generator**: We offload QA functionality to a large, frozen generator LLM, and make another small, trainable search planner LLM focus on search planning, thereby ensuring flexibility for real-world applications.

2) **Dual-Reward Alignment for Search Planning**: We design a dual-reward mechanism to align search planning capabilities at two levels. At the outcome level, an outcome reward quantifies the performance gain of search planning over non-planning baselines (e.g., direct inference or naive RAG). At the process level, a process reward evaluates the rationality of the search planning trajectory.Together, these rewards ensure precise alignment of search planning abilities.

3) **Pareto Optimization of Planning Utility and Cost**: In real-world scenarios, the trade-off between search planning effectiveness (e.g., end-to-end QA accuracy) and computational overhead (e.g., reasoning turns) significantly impacts user experience. We formalize search planning as a pareto optimization problem to jointly maximize planning utility while minimizing planning cost.

To demonstrate the effectiveness of our proposed AI-SearchPlanner, we conduct extensive experiments across multiple search reasoning datasets, and observe that AI-SearchPlanner outperforms existing RL-based search agents in both effectiveness and efficiency, while exhibiting strong generalization capabilities across diverse frozen QA models and data domains.

## 2 RELATED WORK

Although large language models (LLMs) Vaswani et al. (2017); Achiam et al. (2023); Zhao et al. (2023) exhibit remarkable general question-answering capabilities, they still lack the ability to leverage external domain-specific knowledge to solve complex reasoning problems. To mitigate these limitations, search engines are commonly employed to augment LLMs with external information. There are two primary approaches to integrating search engines with LLMs: (1) retrieval-augmented generation (RAG) Lewis et al. (2020); Gao et al. (2023); Mei et al. (2025) and (2) treating the search engine as a tool within an agent framework Schick et al. (2023); Jin et al. (2025); Qu et al. (2025); Song et al. (2025); Wu et al. (2025a); Li et al. (2025a). Retrieval-Augmented Generation (RAG) Lewis et al. (2020); Gao et al. (2023); Mei et al. (2025) enhances LLMs by integrating retrieved external knowledge into the input, thereby providing access to up-to-date or domain-specific information. However, most existing RAG systems Ma et al. (2023); Jiang et al. (2023b;a) follow rigid retrieval-generation pipelines, which lack adaptive planning mechanisms for complex reasoning. As a result, their performance remains heavily dependent on retrieval accuracy, limiting their robustness in dynamic or knowledge-intensive scenarios.

With the advancement of the general capabilities of LLMs, recent research Qu et al. (2025); Trivedi et al. (2022a); Yao et al. (2023); Schick et al. (2023) has explored integrating search engines as tools to enhance problem-solving. Specially, prompt-based methods Trivedi et al. (2022a); Yao et al. (2023) and self-supervised fine-tuning (SFT) approaches Schick et al. (2023) have been proposed to enable LLMs to autonomously determine when and how to invoke search engines during reasoning. For instance, IRCoT Trivedi et al. (2022a) and ReAct Yao et al. (2023) utilize iterative prompting strategies to interleave reasoning with search engine calls, while Toolformer Schick et al. (2023)

---
[2]https://chat.baidu.com/
[3]https://yuanbao.tencent.com/chat/

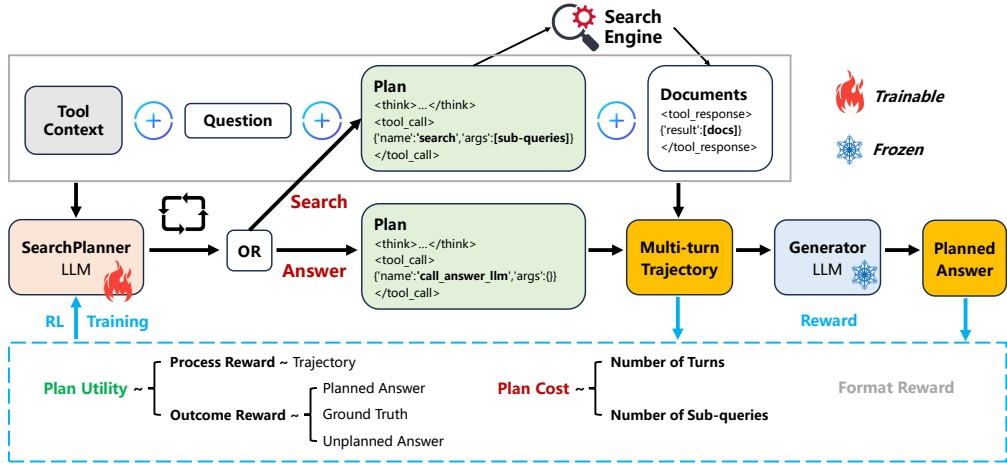

Figure 1: The overview of AI-SearchPlanner framework.

leverages supervised fine-tuning to improve search-related capabilities. However, prompt-based methods often exhibit limited generalization, especially for complex reasoning tasks that lie outside the pretraining distribution of LLMs. SFT-based methods often depend on large-scale, high-quality annotated trajectories of search-reasoning interactions, which limits their scalability. Moreover, recent studies suggest that SFT-based approaches tend to overfit to memorized reasoning paths, further constraining their generalization to novel or out-of-distribution scenarios. Recent studies Jin et al. (2025); Song et al. (2025); Wu et al. (2025a); Li et al. (2025a;c) demonstrate that reinforcement learning (RL) can enable LLMs to develop advanced reasoning capabilities using only outcome-based rewards. For example, Search-R1-like approaches Jin et al. (2025); Song et al. (2025) have investigated the potential of RL methods such as PPO Schulman et al. (2017) and GRPO Shao et al. (2024) in agentic search scenarios. By employing outcome-based reward mechanisms and end-to-end training, these methods enable LLMs to jointly optimize multi-turn search planning and answer generation, achieving strong QA performance. A related concurrent work is s3 Jiang et al. (2025), which also decouples the planner from the generator. Our approach differs from s3 as follows: 1) Objective: s3 aims to train a high-performing planner with minimal data, while we focuses on industrial applications, optimizing the planner for the best utility-cost trade-off using state-of-the-art LLMs. 2) Framework: s3 provides the generator with key documents from the final reasoning step; we instead supply the entire planning trajectory. 3) Method Design: s3 introduces a Gain Beyond RAG reward and filters out training examples solvable by naive RAG, thereby improving end-to-end QA performance on complex questions. We directly optimizes planning ability with an outcome-process dual reward and incorporates Pareto optimization of utility and cost, significantly reducing cost while maintaining performance. Additionally, another similar line of recent work Xiao et al. (2024); Shao et al. (2025), ReasoningIR, enhance IR models with intrinsic reasoning ability, by using LLM reasoning to generate query-document pairs for training. In contrast, we leverages query–search–document planning trajectories to directly produce high-quality answers.

## 3 AI-SEARCHPLANNER

In this section, we introduce AI-SearchPlanner, a novel reinforcement learning framework designed to enhance end-to-end QA performance by focusing on search planning. As illustrated in Figure 1, we demonstrate AI-SearchPlanner framework. Given a question $q$ and its ground truth answer $gt$, a trainable search planner $LLM_{plan}$ iteratively interacts with a search engine $S(\cdot)$: At each turn $t$, $LLM_{plan}$ first generates reasoning context, and then decides to either: 1) Generates sub-queries $\{sq\}^t$ within $\texttt{search}$ tool to retrieve relevant documents via $S(\cdot)$, or 2) Terminates reasoning by generating $\texttt{call\_answer\_llm}$ tool , which feeds the accumulated knowledge from the reasoning trajectory into a frozen generator $LLM_{gen}$ to produce an answer $a$. The final multi-turn reasoning trajectory is represented as $T = \left\{ q, r_1^S, S\left(\{sq\}^1\right), ..., r_L^S, S\left(\{sq\}^L\right), r^A \right\}$, where $r_*^S$ and $r^A$ refer to the planning contexts of searching sub-queries and invoking a frozen generator, respectively. In the following, we introduces three key innovations in AI-SearchPlanner to achieve the objectives.

### 3.1 Decoupling the Architecture of the Search Planner and Generator

In practical applications, advanced AI search systems often rely on large, frozen LLMs to ensure high-quality QA. Thus, to balance effectiveness and efficiency, a more practical approach is to employ a small, trainable LLM dedicated to search planning — enhancing QA performance while maintaining low latency. As shown in Figure 1, our framework offloads QA functionality to a large, frozen generator $LLM_{gen}$, and make another small, trainable search planner $LLM_{plan}$ focus on search planning, thereby ensuring flexibility for real-world applications. Specifically, $LLM_{plan}$ iteratively interacts with the search engine, dynamically deciding whether to generate sub-queries and retrieve relevant documents, or feed the accumulated context from the reasoning trajectory into $LLM_{gen}$ to generate the answer.

### 3.2 Dual-Reward Alignment for Search Planning

Unlike existing methods that rely on a single LLM to jointly handle both search planning and question answering (QA) — aligning end-to-end QA capabilities, we propose a dual-reward mechanism to explicitly align search planning abilities.

- **Outcome Reward** measures the performance gain achieved by search planning compared to non-planning baselines (e.g., direct inference or naive RAG). Let $a$, $a_I$, and $a_R$ denote the answers produced by search planning, direct inference, and naive RAG, respectively. The function $Score\,(\cdot)$ is used to evaluate the consistency between an LLM-generated answer and the corresponding ground truth answer $gt$, with a value within $\in \{0, 1\}$. Specifically, the performance gains of search planning vary across different problems, presenting four different pairs of states, each associated with a specific reward ranking.

$$Rank \left( \left\langle Score\,(a, gt), \max \left\{ \begin{matrix} Score\,(a_I, gt) \\ Score\,(a_R, gt) \end{matrix} \right\rangle \right) \right) \Rightarrow \langle 1, 0 \rangle > \langle 1, 1 \rangle > \langle 0, 0 \rangle > \langle 0, 1 \rangle \quad (1)$$

These above reward rankings incorporates both answer quality and the inherent planning complexity of the questions. The final reward metric can be formalized as follows:

$$R_{outcome} = \frac{1}{2} + Score\,(a, gt) - \frac{1}{2} * \max \left\{ \begin{matrix} Score\,(a_I, gt) \\ Score\,(a_R, gt) \end{matrix} \right. \in [0, 1.5] \quad (2)$$

- **Process Reward** evaluates the rationality of the search planning trajectory $T$ through the frozen generator $LLM_{gen}$ and corresponding prompt $P_T$.

$$R_{process} = LLM_{gen}\,(T, P_T) \in [0, 0.5] \quad (3)$$

**Dual Reward** combines outcome and process rewards to constitute the utility of search planning, ensuring precise alignment with search planning capabilities.

$$R_{utility} = R_{outcome} + R_{process} \quad (4)$$

### 3.3 Pareto Optimization of Planning Utility and Cost

In real-world applications, the trade-off between search planning effectiveness (e.g., end-to-end QA performance) and computational overhead (e.g., search frequency, reasoning turns) critically influences user experience. We formalize search planning as a pareto optimization problem to simultaneously maximize utility while minimizing cost. The overall cost of the planning trajectory $T$ comprises two key components: 1) **The number of planning turns** $L$: Directly impacts inference latency during planning. 2) **The number of sub-queries** $\sum_i^L \left| \{sq\}^i \right|$: Determines the frequency of search engine invocations. We formalize the final planning cost as:

$$R_{cost} = R_{cost}^{turn} + R_{cost}^{query} \quad (5)$$

$$R_{cost}^{turn} = \max \left( 0, 1 - \frac{L}{M_t} \right), \ R_{cost}^{query} = \max \left( 0, 1 - \frac{\sum_i^L \left| \{sq\}^i \right|}{M_q} \right) \quad (6)$$

Where $M_t$ and $M_q$ denote the maximum thresholds for the number of planning turns and sub-queries, respectively. To jointly optimize utility maximization and cost minimization in search planning, we formulate the following Pareto optimization problem.

$$R_{pareto} = R_{utility} + \alpha * R_{cost} + R_{format} \tag{7}$$

Where $\alpha$ is a non-negative coefficient. By adjusting $\alpha$, we guide the search planning towards different trade-offs between utility and cost, enabling exploration of the Pareto frontier. $R_{format}$ denotes the reward for format correctness.

### 3.4 REINFORCEMENT LEARNING TRAINING

We implement a reinforcement learning approach to optimize the search planner's decision-making on when to invoke the search engine or the frozen generator.

- **Reward Design.** We train the search planner to integrate search during inference, enhancing problem-solving accuracy while reducing planning costs. Let $n_S$ and $n_A$ denote the number of search engine and frozen generator invocations, respectively.

$$R = R_{pareto}, \;\; s.t. \;\; R_{format} = \begin{cases} 0, & \text{if format} = \checkmark \;\&\; \text{if } n_S \geq 1 \;\&\; n_A \geq 1 \\ -1, & \text{else} \end{cases} \tag{8}$$

- **Loss Masking for Retrieved Tokens.** During reinforcement learning training, the rollout sequence consists of LLM-generated tokens and retrieved tokens. Since retrieved documents serve as environmental observations rather than LLM-generated content, we apply a loss masking strategy to exclude retrieved tokens from gradient computation. This ensures that the policy gradient objective is computed solely over LLM-generated tokens, preventing interference from retrieved content and preserving the model's intrinsic reasoning and generation capabilities.

- **Optimization of Search Planner.** We optimize the search planning policy using reinforcement learning with a dual reward formulation. Each search planning trajectory comprises an input question, a sequence of retrieved documents, and a termination decision (i.e., call generator to answer). Upon constructing the final context, a frozen generator generates the answer, and the dual reward is computed. In this work, we employ Proximal Policy Optimization (PPO) Schulman et al. (2017). The corresponding PPO objective is defined as:

$$\mathbb{L}(\theta) = \mathbb{E}_{\tau \sim \pi_\theta} \left[ \sum_{t=0}^{T} \min \left( \frac{\pi_\theta(a_t|s_t)}{\pi_{old}(a_t|s_t)} A_t, \text{clip} \left( \frac{\pi_\theta(a_t|s_t)}{\pi_{old}(a_t|s_t)}, 1-\epsilon, 1+\epsilon \right) A_t \right) \right] \tag{9}$$

where $\pi_\theta$ and $\pi_{old}$ denote the current and reference policy models, respectively. $A_t$ corresponds to the estimated advantage. $\epsilon$ is the clipping threshold introduced in PPO to ensure training stability.

## 4 EXPERIMENT

We conduct extensive experiments on multiple benchmark datasets, and validate the effectiveness of our proposed AI-SearchPlanner.

### 4.1 DATASETS

To evaluate AI-SearchPlanner, we use seven Wikipedia-based datasets and two Web-based datasets: 1) **Wiki-based**: NQ Kwiatkowski et al. (2019), TriviaQA Joshi et al. (2017), PopQA Mallen et al. (2022), HotpotQA Yang et al. (2018), 2WikiMultiHopQA Ho et al. (2020), Musique Trivedi et al. (2022b), and Bamboogle Press et al. (2022). Specially, the first three are general QA datasets, and the last four are multi-hop QA datasets. 2) **Web-based**: WebShaper Tao et al. (2025) and WebWalkerQA Wu et al. (2025b).

### 4.2 BASELINES

To evaluate the effectiveness of AI SearchPlanner, we compare it with the following baseline methods that do not decouple the planner and generator: 1) **Inference without Retrieval**: Direct inference and Chain-of-Thought (CoT) reasoning Wei et al. (2022). 2) **Inference with Retrieval**:

Table 1: Performance comparison of all methods on Wikipedia-based datasets. We highlight the best performance among the trainable or frozen models in the same series.

| Method | General QA | | | Multi-Hop QA | | | | Avg. |
|---|---|---|---|---|---|---|---|---|
| | NQ | TriviaQA | PopQA | HotpotQA | 2wiki | Musique | Bamboogle* | |
| **Non-decoupled Planner-Generator** | | | | | | | | |
| ⇒ **Qwen2.5-7b-Instruct** | | | | | | | | |
| ↓ **Planner (Frozen) & Generator (Frozen)** | | | | | | | | |
| Direct Inference | 0.325 | 0.507 | 0.196 | 0.301 | 0.268 | 0.135 | 0.192 | 0.275 |
| Naive RAG | 0.585 | 0.753 | 0.489 | 0.478 | 0.325 | 0.162 | 0.408 | 0.457 |
| CoT | 0.370 | 0.583 | 0.178 | 0.299 | 0.278 | 0.109 | 0.432 | 0.321 |
| IRCoT | 0.537 | 0.703 | 0.442 | 0.394 | 0.194 | 0.105 | 0.336 | 0.387 |
| Search-o1 | 0.462 | 0.652 | 0.350 | 0.441 | 0.400 | 0.198 | 0.472 | 0.425 |
| ↓ **Planner (Trainable) & Generator (Trainable)** | | | | | | | | |
| SFT | 0.556 | 0.553 | 0.177 | 0.384 | 0.269 | 0.166 | 0.400 | 0.358 |
| Search-R1 | **0.630** | **0.771** | 0.503 | 0.571 | 0.402 | 0.215 | 0.544 | 0.519 |
| **Generator Only (Frozen)** | | | | | | | | |
| ⇒ **Qwen3-32b** | | | | | | | | |
| Direct Inference | 0.486 | 0.737 | 0.276 | 0.451 | 0.364 | 0.218 | **0.648** | 0.454 |
| Naive RAG | 0.652 | **0.824** | 0.517 | 0.587 | 0.397 | 0.229 | 0.568 | 0.539 |
| ⇒ **Deepseek-V3** | | | | | | | | |
| Direct Inference | 0.604 | **0.870** | 0.447 | 0.519 | 0.425 | 0.254 | 0.432 | 0.507 |
| Naive RAG | 0.670 | 0.840 | 0.527 | 0.606 | 0.438 | 0.241 | 0.424 | 0.535 |
| ⇒ **Deepseek-R1** | | | | | | | | |
| Direct Inference | 0.657 | **0.903** | 0.499 | 0.655 | 0.594 | 0.365 | 0.632 | 0.615 |
| Naive RAG | 0.668 | **0.903** | 0.495 | 0.661 | 0.599 | 0.369 | 0.648 | 0.620 |
| **Decoupled Planner-Generator** | | | | | | | | |
| ⇒ **AI-SearchPlanner** | | | | | | | | |
| ↓ **Planner (Trainable):** Qwen2.5-7b-Instruct **& Generator (Frozen):** Qwen3-32b | | | | | | | | |
| $\alpha = 0$ | **0.674** | 0.800 | **0.540** | 0.678 | **0.565** | **0.355** | 0.568 | **0.597**(+10.76%) |
| **Generator Transferability** | | | | | | | | |
| Qwen2.5-7b-Instruct | 0.628 | 0.764 | **0.517** | 0.638 | 0.462 | 0.320 | 0.552 | **0.554**(+6.74%) |
| Deepseek-V3 | **0.678** | 0.833 | **0.534** | 0.671 | 0.562 | 0.349 | 0.640 | **0.610**(+14.02%) |
| Deepseek-R1 | **0.719** | 0.880 | **0.549** | 0.741 | 0.617 | 0.373 | 0.656 | **0.648**(+4.52%) |

Naive Retrieval-Augmented Generation (RAG) Lewis et al. (2020), IRCoT Trivedi et al. (2022a), and Search-o1 Li et al. (2025b). 3) **Training-based Methods**: Supervised fine-tuning (SFT) Chung et al. (2024) and Search-R1 Jin et al. (2025).

## 4.3 EXPERIMENTAL SETTINGS

For training AI-SearchPlanner, We merge the training sets of NQ and HotpotQA to form a unified dataset. Qwen-2.5-7b-Instruct[4] and Qwen3-32b[5] were used as the trainable planner and frozen generator, respectively. The maximum thresholds for the number of reasoning turns and sub-queries $M_t$ and $M_q$ is set to 5 and 10, respectively. The prompts for evaluating process reward and answer accuracy are provided in Appendix B. Since different frozen QA models may produce answers with divergent content and stylistic variations, the Exact Match (EM) metric is inadequate for measuring answer correctness. To address this limitation, we introduce an LLM-based scoring function $Score(\cdot)$ that evaluates the answer accuracy of model responses based on the question and the ground truth answer. On the Wikipedia-based datasets, we follow the settings in Jin et al. (2025) to conduct the experiments. We use the 2018 Wikipedia dump Karpukhin et al. (2020) as the knowledge source and E5 Wang et al. (2022) as the retriever. The number of retrieved passages is set to 3. The maximum number of reasoning turns is set to 5. On the Web-based datasets, we use Google search API[6] as the knowledge retriever. The number of retrieved titles and snippets is set to 10. The maximum number of reasoning turns is set to 10. We provide the case study in Appendix A.

---

[4]https://huggingface.co/Qwen/Qwen2.5-7B-Instruct

[5]https://huggingface.co/Qwen/Qwen3-32B

[6]https://serper.dev/

Table 2: Performance Comparison of AI-Searchplanner transferred to Web-based data domains.

| Method | Web QA | |
|---|---|---|
| | WebShaper | WebWalker |
| ⇒ **Qwen2.5-7b-Instruct** | | |
| Direct Inference | 0.110 | 0.054 |
| Naive RAG | 0.188 | 0.297 |
| CoT | 0.104 | 0.022 |
| IRCoT | 0.140 | 0.228 |
| Search-o1 | 0.186 | 0.231 |
| ⇒ **Qwen3-32b** | | |
| Direct Inference | 0.170 | 0.057 |
| Naive RAG | 0.204 | 0.299 |
| ⇒ **AI-SearchPlanner** | | |
| ↓ **Planner (Trainable)**: Qwen2.5-7b-Instruct **& Generator (Frozen)**: Qwen3-32b | | |
| $\alpha = 0$ | **0.366**(+79.41%) | **0.375**(+25.42%) |
| **Generator Transferability** | | |
| Qwen2.5-7b-Instruct | **0.322**(+71.28%) | **0.306**(+3.03%) |

## 4.4 RESULTS

In this section, we demonstrate our experimental results, and provide detailed analysis.

### 4.4.1 OVERALL PERFORMANCE

The experimental results on Wikipedia-based datasets are given in Table 1. For the overall performance, AI-SearchPlanner state-of-the-art performance, substantially outperforming all baseline methods. In detail, in terms of **Search Planning Ability**, the results indicate that: (1) When employing the same frozen generator, AI-SearchPlanner yields significant performance gains over both non-planning baselines (e.g., direct inference and naive RAG) and existing planning-based methods; (2) Notably, the model exhibits more pronounced improvements on multi-hop questions compared to single-hop questions. This indicates the capability of dual reward-aligned search planning mechanism optimized via reinforcement learning, highlighting its superior reasoning and retrieval strategies in addressing complex questions. In terms of **Architecture Decoupling**, we can find that: When transferring to a larger frozen generator, AI-SearchPlanner maintains consistent performance improvements over both generator-only baselines and non-decoupled architectures. This result validates the advantage of decoupling the planner module from the generator, as it allows the planner to specialize explicitly in search planning while facilitating seamless integration with more powerful generators. Consequently, this flexibility leads to robust performance enhancements without requiring model retraining. Additionally, on the Wiki dataset, we observe that the performance gain of naive RAG over direct inference diminishes with more powerful generators. This suggests that stronger generators may have memorized more comprehensive wiki knowledge during pre-training.

### 4.4.2 DOMAIN TRANSFERABILITY

In Section 4.4.1, we have demonstrated the transferability of AI-SearchPlanner across different frozen generators. In this section, we discuss the transferability of the wiki-trained AI-SearchPlanner on web-based datasets and search engines. As shown in Table 2, AI-SearchPlanner consistently outperforms baseline methods in overall answer accuracy, even when adapted to out-of-domain datasets and search engines. These results demonstrate the robustness and strong transferability of the learned search planning capability. Moreover, on more complex web-based datasets, untrained planning baselines performs even worse than no-planning baselines. This indicates that oversimplified search planning strategies can be detrimental to solving challenging reasoning problems.

### 4.4.3 TRAINING DYNAMICS

In Figure 2, we present the key dynamics during the training process of AI-SearchPlanner with cost coefficient $\alpha = 0$. For **Reward-related Dynamics**, the first row of Figure 2 illustrates three key rewards influencing training performance: LLM-based answer reward, process reward, and format reward. As training progresses, all three rewards exhibit consistent improvement, with the format

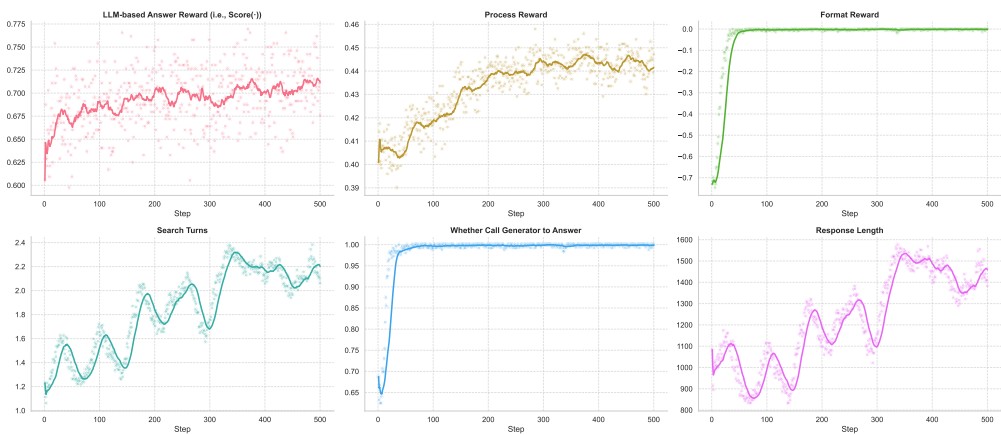

Figure 2: Training dynamics of AI-SearchPlanner with cost coefficient $\alpha = 0$.

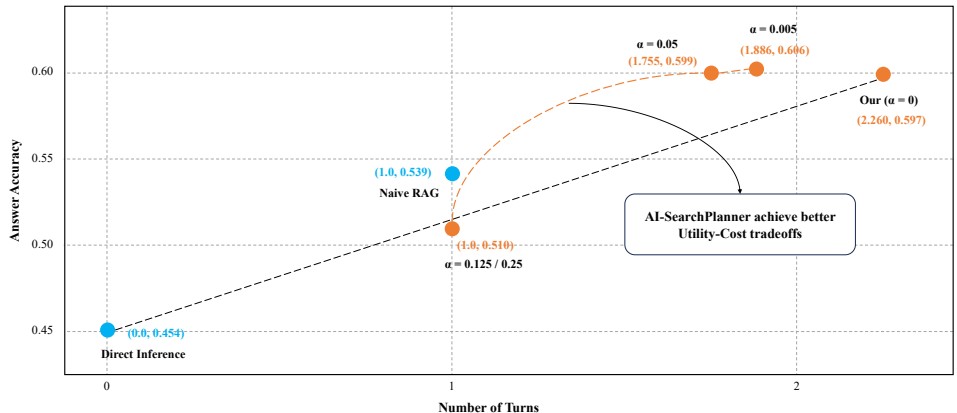

Figure 3: Utility-Cost tradeoffs on Wikipedia-based datasets. Blue points represent non-planning baselines. Orange points represent AI-SearchPlanner with differet cost coefficient $\alpha$.

reward converging significantly faster than the other two. This suggests that AI-SearchPlanner effectively enhances both the quality of generated answers and the rationality of reasoning trajectories, while format correctness is learned earlier in the training process. For **Action-related Dynamics**, the second row of Figure 2 presents three important actions during the training process: number of turns calling the search engine, whether the frozen generator is invoked, and response length. We observe that, as training steps increase: 1) AI-SearchPlanner learns to call the search engine more frequently to solve problems, resulting in longer responses; 2) After obtaining sufficient external knowledge from multi-turn searches, AI-SearchPlanner quickly learns to terminate reasoning and invoke the frozen generator to produce answers. This demonstrates that during training, AI-SearchPlanner learns how to utilize the search engine and the frozen generator to address complex reasoning problems.

### 4.4.4 PARETO FRONTIER ANALYSIS OF UTILITY-COST

We trained four variants of AI-SearchPlanner by varying the cost coefficient $\alpha \in \{0.005, 0.05, 0.125, 0.25\}$. The experimental results are given in Figure 3 and Table 3, we can find that, AI-SearchPlanner achieves a compelling Pareto frontier compared to non-plannling baselines. Specifically, as the cost coefficient $\alpha$ increases, the number of search reasoning turns employed by AI-SearchPlanner continuously decreases, eventually converging to 1, while the answer performance initially exhibits a slight improvement before undergoing a consistent decline. This highlights AI-SearchPlanner's effectiveness in navigating the trade-off between performance and cost.

Table 3: Performance and cost comparison of AI-SearchPlanner with different cost coefficient $\alpha$ on Wikipedia-based Datasets. The numbers in [] represent the search planning turns. We highlight the best performance among these models.

| Method | General QA | | | Multi-Hop QA | | | | Avg. |
|---|---|---|---|---|---|---|---|---|
| | NQ | TriviaQA | PopQA | HotpotQA | 2wiki | Musique | Bamboogle | |
| $\Rightarrow$ AI-SearchPlanner | | | | | | | | |
| $\alpha = 0$ | **0.674** [1.856] | 0.800 [1.910] | 0.540 [1.749] | **0.678** [2.352] | 0.565 [2.681] | 0.355 [2.867] | 0.568 [2.408] | 0.597 [2.260] |
| $\alpha = 0.005$ | 0.669 [[1.388] | **0.809** [1.304] | 0.546 [1.546] | 0.677 [1.961] | **0.570** [2.452] | **0.368** [2.476] | 0.600 [2.072] | **0.606**(+1.51%) [1.886](-16.55%) |
| $\alpha = 0.05$ | 0.671 [1.135] | 0.798 [1.157] | 0.543 [1.248] | 0.657 [1.858] | 0.535 [2.313] | 0.346 [2.596] | **0.640** [1.976] | 0.599(+0.34%) [1.755](-22.35%) |
| $\alpha = 0.125$ | 0.662 [1.000] | 0.795 [1.000] | 0.542 [1.000] | 0.564 [1.000] | 0.385 [1.000] | 0.219 [1.000] | 0.400 [1.000] | 0.510(-14.57%) [1.000](-55.75%) |
| $\alpha = 0.25$ | 0.650 [1.000] | 0.797 [1.000] | **0.564** [1.000] | 0.560 [1.000] | 0.380 [1.000] | 0.204 [1.000] | 0.416 [1.000] | 0.510(-14.57%) [1.000](-55.75%) |

Table 4: Performance comparison of all ablation models on Wikipedia-based datasets. We highlight the best performance among the ablation models.

| Method | General QA | | | Multi-Hop QA | | | | Avg. |
|---|---|---|---|---|---|---|---|---|
| | NQ | TriviaQA | PopQA | HotpotQA | 2wiki | Musique | Bamboogle | |
| $\Rightarrow$ AI-SearchPlanner | | | | | | | | |
| $\alpha = 0$ | 0.674 | 0.800 | 0.540 | **0.678** | **0.565** | **0.355** | **0.568** | **0.597** |
| w/o Outcome Gain | 0.644 | 0.796 | **0.557** | 0.557 | 0.375 | 0.200 | 0.416 | 0.506(-15.24%) |
| w/o Process Reward | **0.686** | **0.813** | **0.557** | 0.674 | 0.529 | 0.327 | 0.528 | 0.588(-1.51%) |
| w/o RL Training | 0.616 | 0.772 | 0.447 | 0.623 | 0.528 | 0.325 | 0.520 | 0.547(-8.38%) |

### 4.4.5 ABLATION STUDY

In Table 4, we analyze the effects of the dual reward mechanism and whether to train the planner. For **Dual-Reward Mechanism**, we remove one of them once a time to analyze its contribution. As shown in Table 4, we can observe that, 1) Removing outcome gain (i.e., $Score\,(\cdot)$ only) or process reward in AI-SearchPlanner will bring a decline in performance; 2) Compared to the ablation baselines, AI-SearchPlanner demonstrates significant performance improvements on complex multi-hop problems, which proves the effectiveness of dual-reward alignment. For **Trainable vs. Frozen Planner**, we can find that: Compared to the frozen planner, the trainable planner demonstrates a significant improvement in answer quality, which demonstrates the capability of reinforcement learning training.

## 5 CONCLUSION & FUTURE WORK

In this paper, we propose AI-SearchPlanner, a novel reinforcement learning (RL) framework designed to enhance the effectiveness of frozen QA models through optimized search planning. By decoupling the search planner from the generator, our approach enables a small, trainable LLM to specialize in search planning while leveraging a large, frozen LLM for high-quality QA —— an architecture that aligns with real-world AI search system constraints. The framework introduces three key innovations: (1) Decoupling the architecture of the search planner and generator, (2) Dual-reward alignment to refine search planning capabilities, and (3) Pareto optimization to balance planning utility and cost. Extensive experiments on real-world datasets demonstrate that AI-SearchPlanner achieves superior or comparable effectiveness to existing RL-based search agents while maintaining higher efficiency. Our findings underscore the advantages of separating search planning from QA generation, enabling more scalable and adaptive search-reasoning systems. Future work may explore extending this framework to multi-modal search tasks or integrating dynamic reward mechanisms for broader generalization. Ultimately, AI-SearchPlanner offers a practical and efficient solution for enhancing LLM-powered search systems, advancing the development of more intelligent and resource-efficient AI assistants.

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

# A    CASE STUDY

We present two case studies to demonstrate the performance of AI-Searchplanner in solving complex problems. In both cases, AI-Searchplanner arrives at the correct solution after multiple ($\geq 5$) rounds of search and planning. In contrast, non-planning baselines fails to reach a correct solution.

**\<Question\>**: In what county is the city that shares a border with the capital of the state where Levi Casey was born?
**\<Ground Truth\>**: ['Richland County', 'Richland County, South Carolina']

**Qwen3-32B: Direct Inference** ✗

**\<Response\>**: Grundy County

**Qwen3-32B: Naive RAG** ✗

**\<Context\>**: Doc 1: Idaho Idaho () is a state in the northwestern region of the United States. It borders the state of Montana to the east and northeast, Wyoming to the east, Nevada and Utah to the south, and Washington and Oregon to the west. To the north, it shares a small portion of the Canadian border with the province of British Columbia. With a population of approximately 1.7 million and an area of , Idaho is the 14th largest, the 12th least populous and the 7th least densely populated of the 50 U.S. states. The state's capital and largest city is Boise.
Doc 2: Lyndon, Kentucky Lyndon is a home rule-class city in Jefferson County, Kentucky, United States. The population was 11,002 at the 2010 census, up from 9,369 at the 2000 census. Incorporated on May 10, 1965, Lyndon became part of the new Louisville Metro government in 2003. It remains an independent city with its own mayor and emergency services and is not counted in Louisville's population although its citizens can vote for the mayor of Louisville and Metro Council members. The area is the home of Oxmoor Farm, the residence of Alexander Scott Bullitt, one of the drafters of Kentucky's first constitution.
Doc 3: Roman Catholic Diocese of Davenport The Roman Catholic Diocese of Davenport () is a diocese of the Catholic Church for the southeastern quarter of the U.S. state of Iowa. There are within the diocese. The diocese's eastern border is at the Mississippi River; the northern border comprises the counties of Jasper, Poweshiek, Iowa, Johnson, Cedar, and Clinton; the western border is made up of the counties of Jasper, Marion, Monroe, and Appanoose; and the southern border is the Iowa-Missouri border. The current bishop of the diocese is Bishop Thomas Zinkula. It is a suffragan see of the Archdiocese of Dubuque.
**\<Response\>**: Jefferson County

**AI-SearchPlanner** ✓

**\<Planning Trajectory\>**: To answer the question, I need to find out the following information step by step:
1.  Find out where Levi Casey was born.
2.  Determine the state where Levi Casey was born.
3.  Find out the capital of that state.
4.  Find out the city that shares a border with the capital of that state.
5.  Find out the county where that city is located.
I will start with finding out where Levi Casey was born.
\<tool_call\>
{"name": "search", "arguments": {"query_list"}: ["where was Levi Casey born"]}}
\</tool_call\>
\<tool_response\>
{"result": "Search query:where was Levi Casey born.
Doc 1 (Title: "Levi Casey (politician)") Levi Casey (politician) General Levi Casey (c. 1752 { February 3, 1807) was a United States Representative from South Carolina. He was born in South Carolina and served in the South Carolina militia during the American Revolutionary War. Son of Abner Casey, he served in the American Revolutionary War along with his step brothers, brother(s) and nephews. One of his stepbrothers, Benjamin Casey was killed in action at Camp Middlebrook, New Jersey. He married Elizabeth Duckett in 1775. He reached the rank of brigadier general in the South Carolina militia. After the war, he served as justice of Newberry County.
Doc 2 (Title: "Lyman R. Casey") a State into the Union, Casey was elected as a Republican to the U.S. Senate and served from November 25, 1889, to March 4, 1893. He was an unsuccessful candidate for renomination in 1892. While in the Senate, he was chairman of the Committee on Railroads (Fifty-second Congress). He moved to New York City. In 1861, Casey married Harriett Mary Platt. One of her siblings was Jane Platt, wife of the drinking straw inventor Marvin C. Stone. Casey returned to Washington, D.C.. On January 25, 1914, Casey died in Washington DC. Casey is buried at Greenmount Cemetery in Baltimore, Maryland.
Doc 3 (Title: "Levi Greenwood") 2012 !  scope="row" style="text-align:center" | 2013 !  scope="row" style="text-align:center" | 2014 !  scope="row" style="text-align:center" | 2015 !  scope="row" style="text-align:center" | 2016 !  scope="row" style="text-align:center" | 2017 !  scope="row" style="text-align:center" | 2018 !  colspan=3| Career !  135 !  56 !  63 !  1322 !  1028 !  2350 !  444 !  614 !  0.4 !  0.5 !  9.8 !  7.6 !  17.4 !  3.3 !  4.6 Levi Greenwood Levi Greenwood (born 19 February 1989) is an Australian rules footballer, currently playing for the Collingwood Football Club in the Australian Football League (AFL). Greenwood grew up in the small coastal town of Port.  }
\</tool_response\>
Doc 1 states that Levi Casey (the politician) was born in South Carolina. I will proceed with the next step.
\<tool_call\>
{"name": "search", "arguments": {"query_list"}: ["capital of the state where Levi Casey was born"]}}
\</tool_call\>
\<tool_response\>
{"result": "Search query:capital of the state where Levi Casey was born.

```
Doc 1 (Title: "Levi Casey (politician)") Levi Casey (politician) General Levi Casey (c.  1752 { February 3,
1807) was a United States Representative from South Carolina.  He was born in South Carolina and served in the
South Carolina militia during the American Revolutionary War.  Son of Abner Casey, he served in the American
Revolutionary War along with his step brothers, brother(s) and nephews.  One of his stepbrothers, Benjamin
Casey was killed in action at Camp Middlebrook, New Jersey.  He married Elizabeth Duckett in 1775.  He reached
the rank of brigadier general in the South Carolina militia.  After the war, he served as justice of Newberry
County.  Doc 2 (Title:  "Zadok Casey") in the Illinois House from 1848 to 1852, serving as speaker in 1852,
and in the State Senate from 1860 to 1862.  Casey Creek, a tributary of the Big Muddy River, is named in
honor of Zadok Casey.  Casey Middle School and Casey Avenue in Mount Vernon are also named after him.  Zadok
Casey was born in Greene County, Georgia.  Not much is known about his early life.  One story is that, as a
young man, he witnessed a murder.  Because he did not wish to testify, he fled to the frontier.  He died in
Caseyville, Illinois at age 66,
Doc 3 (Title: "Levi Casey (politician)") Court in 1785.  Casey was a member of the South Carolina Senate in
1781 and 1782 and 1800{1802 and a member of the South Carolina House of Representatives 1786{1788, 1792{1795
and 1798{1799.  He was elected as a Republican to the Eighth and Ninth Congresses and served from March 4,
1803, until his death, before the close of the Ninth Congress.  Prior to dying, he had been reelected to the
Tenth Congress.  He died in Washington, D.C., February 3, 1807 and was buried in the Congressional Cemetery.
Levi Casey (politician) General Levi Casey (c.  1752 { February 3, 1807) was a"}
</tool_response>
Doc 1 confirms that Levi Casey (the politician) was born in South Carolina.  Now I will proceed with finding
the capital of South Carolina.
<tool_call>
{"name":  "search", "arguments":  {"query_list"}:  ["capital of South Carolina"]}}
</tool_call>
<tool_response>
{"result":  "Search query:capital of South Carolina.
Doc 1 (Title:  "South Carolina") 40th most extensive and 23rd most populous U.S. state.  Its GDP as of 2013
was $183.6 billion, with an annual growth rate of 3.13%.  South Carolina is composed of 46 counties.  The
capital is Columbia with a 2016 population of 134,309; while its largest city is Charleston with a 2016
population of 134,385.  The Greenville-Anderson-Mauldin metropolitan area is the largest in the state, with
a 2016 population estimate of 884,975.  South Carolina is named in honor of King Charles I of England, who
first formed the English colony, with "Carolus" being Latin for "Charles".  South Carolina is known for its.
Doc 2 (Title:  "Columbia, South Carolina") Columbia, South Carolina Columbia is the capital and second largest
city of the U.S. state of South Carolina, with a population estimate of 134,309 as of 2016.  The city serves
as the county seat of Richland County, and a portion of the city extends into neighboring Lexington County.
It is the center of the Columbia metropolitan statistical area, which had a population of 767,598 as of the
2010 United States Census, growing to 817,488 by July 1, 2016, according to 2015 U.S. Census estimates.  The
name "Columbia" is a poetic term used for the United States, originating from the name.
Doc 3 (Title: "South Carolina State House") South Carolina State House The South Carolina State House is the
building housing the government of the U.S. state of South Carolina.  The building houses the South Carolina
General Assembly and the offices of the Governor and Lieutenant Governor of South Carolina.  Until 1971, it
also housed the Supreme Court.  It is located in the capital city of Columbia near the corner of Gervais and
Assembly Streets.  The State House is in the Greek Revival style; it is approximately tall, long, wide.  It
weighs more than and has of space.  The old State House was constructed between 1786 and 1790."}
</tool_response>
Doc 1 states that the capital of South Carolina is Columbia.  Now I will proceed with finding the city that
shares a border with the capital of South Carolina.
<tool_call>
{"name":  "search", "arguments":  {"query_list"}:  ["city that shares a border with Columbia, South
Carolina"]}}
</tool_call>
<tool_response>
{"result":  "Search query:city that shares a border with Columbia, South Carolina.
Doc 1 (Title:  "Columbia, South Carolina") The city of Columbia has 5 sister cities:  Columbia, South Carolina
Columbia is the capital and second largest city of the U.S. state of South Carolina, with a population
estimate of 134,309 as of 2016.  The city serves as the county seat of Richland County, and a portion of the
city extends into neighboring Lexington County.  It is the center of the Columbia metropolitan statistical
area, which had a population of 767,598 as of the 2010 United States Census, growing to 817,488 by July 1,
2016, according to 2015 U.S. Census estimates.  The name "Columbia" is a poetic term used.
Doc 2 (Title:  "Columbia, South Carolina") Columbia, South Carolina Columbia is the capital and second largest
city of the U.S. state of South Carolina, with a population estimate of 134,309 as of 2016.  The city serves
as the county seat of Richland County, and a portion of the city extends into neighboring Lexington County.
It is the center of the Columbia metropolitan statistical area, which had a population of 767,598 as of the
2010 United States Census, growing to 817,488 by July 1, 2016, according to 2015 U.S. Census estimates.  The
name "Columbia" is a poetic term used for the United States, originating from the name.
Doc 3 (Title:  "Columbia, South Carolina") of Christopher Columbus.  The city is located approximately
northwest of the geographic center of South Carolina, and is the primary city of the Midlands region of the
state.  It lies at the confluence of the Saluda River and the Broad River, which merge at Columbia to form the
Congaree River.  Columbia is home to the University of South Carolina, the state's flagship university and the
largest in the state, and is also the site of Fort Jackson, the largest United States Army installation for
Basic Combat Training.  Columbia is also located 20 miles west of the site of McEntire Joint"}
</tool_response>
Doc 1 states that Columbia, South Carolina, is the capital and serves as the county seat of Richland County.
Therefore, the city that shares a border with Columbia, South Carolina, is in Richland County.  Now I will
proceed with the final step to find out the county where the city that shares a border with the capital of the
state where Levi Casey was born is located.
<tool_call> {"name":  "search", "arguments":  {"query_list"}:  ["county that contains Columbia, South
Carolina"]}}
</tool_call>
<tool_response>
{"result":  "Search query:county that contains Columbia, South Carolina.
Doc 1 (Title:  "Columbia, South Carolina") Columbia, South Carolina Columbia is the capital and second largest
city of the U.S. state of South Carolina, with a population estimate of 134,309 as of 2016.  The city serves
as the county seat of Richland County, and a portion of the city extends into neighboring Lexington County.
It is the center of the Columbia metropolitan statistical area, which had a population of 767,598 as of the
2010 United States Census, growing to 817,488 by July 1, 2016, according to 2015 U.S. Census estimates.  The
name "Columbia" is a poetic term used for the United States, originating from the name.
```

```
Doc 2 (Title:  "Richland County, South Carolina") Richland County, South Carolina Richland County is located
in the U.S. state of South Carolina.  As of the 2016 census estimate, the population was 409,549, making
it the second-most populous county in South Carolina, behind only Greenville County.  The county seat and
largest city is Columbia, the state capital.  The county was founded in 1785.  Richland County is part of the
Columbia, SC Metropolitan Statistical Area.  In 2010, the center of population of South Carolina was located
in Richland County, in the city of Columbia.  Richland County was probably named for its "rich land." The
county was formed in 1785.
Doc 3 (Title:  "Columbia, South Carolina") The city of Columbia has 5 sister cities:  Columbia, South Carolina
Columbia is the capital and second largest city of the U.S. state of South Carolina, with a population
estimate of 134,309 as of 2016.  The city serves as the county seat of Richland County, and a portion of the
city extends into neighboring Lexington County.  It is the center of the Columbia metropolitan statistical
area, which had a population of 767,598 as of the 2010 United States Census, growing to 817,488 by July 1,
2016, according to 2015 U.S. Census estimates.  The name "Columbia" is a poetic term used"}
</tool_response>
Doc 1 states that Columbia, South Carolina, is the county seat of Richland County.  Now I have enough
information to call answer expert llm to generate the final answer.
<tool_call>
{"name":  "call_answer_llm", "arguments":  {}}
</tool_call>
<Response>: Richland County
```

---

**<Question>:**  Storm of the Century filmed in state A based on the book by the author who wrote a
story featuring the author of The Hope.  When do alcohol sales start in state A?
**<Ground Truth>:** ['5am']

**Qwen3-32B: Direct Inference** ✗

**<Response>:** 9:00 AM

**Qwen3-32B: Naive RAG** ✗

**<Context>:**  Doc 1:  Miracle (novel) Miracle is a novel written by Danielle Steel and published by Random House
in June 2005.  The book is Steel's sixty-sixth novel.  It is New Year's Eve when the storm of the century hits
northern California.  In a quiet neighborhood in San Francisco, amid the chaos of fallen trees and damaged
homes, the lives of three strangers are about to collide.  For Quinn Thompson, what happens in the storm's
wake will bring down a barrier he has built around himself since his wife's death.  For neighbor Maggie
Dartman, it will spark friendship at a time when she
Doc 2:  Dry state A dry state is a state in the United States in which the manufacture, distribution,
importation, and sale of alcoholic beverages are prohibited or tightly restricted.  While some states,
such as North Dakota, entered the United States as dry states, others went dry after passage of prohibition
legislation.  Currently, no state in the United States remains completely dry, but some states contain dry
counties.  Prior to the adoption of nationwide prohibition in 1920, state legislatures in the United States
passed local option laws that allowed a county or township to go dry if it chose to do so.
Doc 3:  Rick Moody Hiram Frederick ""Rick"" Moody III (born October 18, 1961) is an American novelist and
short story writer best known for the 1994 novel ""The Ice Storm"", a chronicle of the dissolution of two
suburban Connecticut families over Thanksgiving weekend in 1973, which brought him widespread acclaim, became
a bestseller, and was made into the film ""The Ice Storm"".  Many of his works have been praised by fellow
writers and critics alike, and in 1999 ""The New Yorker"" chose him as one of America's most talented young
writers, placing him on their ""20 Writers for the 21st Century"" list.
**<Response>:** The question cannot be answered with the given context.

**AI-SearchPlanner** ✓

**<Planning Trajectory>:**    To answer the question, I need to find out the state A where the movie "Storm of the
Century" was filmed, the author of the book it was based on, and the author of "The Hope".  Then I will find
out when alcohol sales start in state A. I will start with finding out the state A where the movie "Storm of
the Century" was filmed.
<tool_call>
{"name":  "search", "arguments":  {"query_list"}:  ["state where the movie Storm of the Century was filmed"]}}
</tool_call>
<tool_response>
{"result":  "Search query:state where the movie Storm of the Century was filmed.
Doc 1 (Title:  "Storm of the Century") Storm of the Century Storm of the Century, alternatively known as
Stephen King's Storm of the Century, is a 1999 horror TV miniseries written by Stephen King and directed by
Craig R. Baxley.  Unlike many other King mini-series, "Storm of the Century" was not based upon a Stephen King
novel|King wrote it as a screenplay from the beginning.  The screenplay was published in February 1999.  A
very powerful blizzard hits the fictional small town of Little Tall Island (also the setting of King's novel
"Dolores Claiborne") off the coast of Maine.  The storm is so powerful that all access off
Doc 2 (Title:  "1993 Storm of the Century") north-west Florida early on the morning of March 13, with a
significant storm surge in the northwestern Florida peninsula that drowned several people.  Barometric
pressures recorded during the storm were low.  Readings of were recorded in Tallahassee, Florida, and even
lower readings of were observed in New England.  Low pressure records for March were set in areas of twelve
states along the Eastern Seaboard, with all-time low pressure records set between Tallahassee and Washington,
D.C. Snow began to spread over the eastern United States, and a large squall line moved from the Gulf of
Mexico into Florida and Cuba.

```
Doc 3 (Title: "1993 Storm of the Century") Heavy snow was first reported in highland areas as far south
as Alabama and northern Georgia, with Union County, Georgia reporting up to of snow in the north Georgia
mountains. Birmingham, Alabama, reported a rare of snow. The Florida Panhandle reported up to of snow,
with hurricane-force wind gusts and record low barometric pressures. Between Louisiana and Cuba, the
hurricane-force winds produced high storm surges across the big bend of Florida which, in combination with
scattered tornadoes, killed dozens of people. Record cold temperatures were seen across portions of the south
and east of the US in the wake of"}
</tool_response>
Doc 1 states that "Storm of the Century" is a 1999 horror TV miniseries written by Stephen King and directed
by Craig R. Baxley. It was filmed in Maine, as it is based on a fictional small town of Little Tall Island
off the coast of Maine. Now I will proceed with finding out the author of the book it was based on.
<tool_call>
{"name": "search", "arguments": {"query_list": ["author of the book Storm of the Century by Stephen
King"]}}
</tool_call>
<tool_response>
{"result": "Search query:author of the book Storm of the Century by Stephen King.
Doc 1 (Title: "Storm of the Century") Storm of the Century Storm of the Century, alternatively known as
Stephen King's Storm of the Century, is a 1999 horror TV miniseries written by Stephen King and directed by
Craig R. Baxley. Unlike many other King mini-series, "Storm of the Century" was not based upon a Stephen King
novel|King wrote it as a screenplay from the beginning. The screenplay was published in February 1999. A
very powerful blizzard hits the fictional small town of Little Tall Island (also the setting of King's novel
"Dolores Claiborne") off the coast of Maine. The storm is so powerful that all access off.
Doc 2 (Title: "Storm of the Century") the format the story should take, either a novel or a screenplay,
ending up writing a "novel for television". "Storm of the Century" aired on ABC on February 14{18, 1999.
It was released on DVD on June 22 the same year. The miniseries was well received by critics. Review
aggregator Rotten Tomatoes counts six positive out of eight reviews. U.S. Television Ratings: Storm of the
Century Storm of the Century, alternatively known as Stephen King's Storm of the Century, is a 1999 horror TV
miniseries written by Stephen King and directed by Craig R. Baxley. Unlike many other King mini-series.
Doc 3 (Title: "Storm of the Century") the wrong decision, "but in daylight, I know better." The screenplay
for the miniseries was written by Stephen King expressly for television, and was published by Pocket Books
just prior to the initial airing of "Storm of the Century" on ABC, while the mass market edition of the
screenplay was published only as a trade paperback. A hardcover edition was published concurrently by the
Book of the Month Club. The book contains an introduction in which King describes the genesis of the idea as
it occurred to him in late 1996, beginning to write it in December 1996, and debating"}
</tool_response>
Doc 1 states that "Storm of the Century" is a 1999 horror TV miniseries written by Stephen King. Now I will
proceed with finding out the author of the book "The Hope".
<tool_call>
{"name": "search", "arguments": {"query_list": ["author of the book The Hope"]}}
</tool_call>
<tool_response>
{"result": "Search query:author of the book The Hope.
Doc 1 (Title: "John Hope Bryant") John Hope Bryant John Hope Bryant (born February 6, 1966) is an American
financial literacy entrepreneur and businessman. Bryant is the founder, chairman and chief executive officer
of nonprofit Operation HOPE, chief executive officer of Bryant Group Ventures and The Promise Homes Company,
co-founder of Global Dignity, advisor to business and government and author of bestselling books on economics
and leadership. He served as a member of the President's Advisory Council on Financial Capability, the vice
chair on the U.S. President's Advisory Council on Financial Literacy and as the chairman of the Committee on
the Underserved. Bryant was born on.
Doc 2 (Title: "The Life of Hope") The Life of Hope The Life of Hope is a novel by Paul Quarrington, published
in 1985 by Doubleday Canada. It is part of an unofficial trilogy with Quarrington's later novels "King
Leary" and "Logan in Overtime"; although none of the novels centre on the same protagonists, they all feature
some background interrelationships of character and setting. The novel's central character, essentially an
authorial self-insertion, is a novelist named Paul who is suffering from writer's block after the publication
of his baseball-themed novel "Home Game". Attending a writer's retreat in the small Southern Ontario town of
Hope, he learns about.
Doc 3 (Title: "The Hope (novel)") The Hope (novel) The Hope is a historical novel by Herman Wouk about
pivotal events in the history of the State of Israel from 1948 to 1967. These include Israel's War of
Independence, the 1956 Sinai War (known in Israel as "Operation Kadesh"), and the Six-Day War. The narrative
is continued in the sequel "The Glory". In a loose way it can be considered a sequel to Wouk's Second World
War books – "The Winds of War" and "War and Remembrance". Though the characters are different, the earlier
books gave a major part to the Holocaust and several to their"}
</tool_response>
Doc 1 states that "The Hope" is a book by Herman Wouk. Now I will proceed with finding out the author of the
book "Dolores Claiborne" by Stephen King, as it is mentioned that "Storm of the Century" is based on a novel
by Stephen King.
<tool_call>
{"name": "search", "arguments": {"query_list": ["author of the book Dolores Claiborne by Stephen King"]}}
</tool_call>
<tool_response>
{"result": "Search query:author of the book Dolores Claiborne by Stephen King.
Doc 1 (Title: "Dolores Claiborne") Dolores Claiborne Dolores Claiborne () is a 1992 psychological thriller
novel by Stephen King. The novel is narrated by the title character. Atypically for a King novel, it has
no chapters, double-spacing between paragraphs, or other section breaks; thus the text is a single continuous
narrative which reads like the transcription of a spoken monologue. It was the best-selling novel of 1992 in
the United States. The book is dedicated to King's mother: "For my mother, Ruth Pillsbury King." Dolores
Claiborne, an opinionated 65-year-old widow living on the tiny Maine community of Little Tall Island, is
suspected of murdering her.
Doc 2 (Title: "Dolores Claiborne") Dolores, with Jennifer Jason Leigh as her daughter Selena, and Judy
Parfitt as Vera Donovan. "Dolores Claiborne", the operatic adaptation of the novel composed by Tobias Picker
to a libretto by J. D. McClatchy, premiered at San Francisco Opera on September 18, 2013, with Patricia
Racette in the title role. Dolores Claiborne Dolores Claiborne () is a 1992 psychological thriller novel by
Stephen King. The novel is narrated by the title character. Atypically for a King novel, it has no chapters,
double-spacing between paragraphs, or other section breaks; thus the text is a single continuous narrative
which reads like the.
Doc 3 (Title: "Dolores Claiborne (film)") Dolores Claiborne (film) Dolores Claiborne is a 1995 American
psychological thriller drama film directed by Taylor Hackford and starring Kathy Bates, Jennifer Jason Leigh,
and David Strathairn. It is based on the novel of the same name by Stephen King. The plot focuses on the
strained relationship between a mother and her daughter, largely told through flashbacks, after her daughter
arrives to her remote hometown on a Maine island where her mother has been accused of murdering the elderly
woman for whom she cared. The screenplay for "Dolores Claiborne" was adapted by Tony Gilroy, and the film was
shot in"}
</tool_response>
```

```
Doc 1 states that "Dolores Claiborne" is a 1992 psychological thriller novel by Stephen King.  Now I have
enough information to proceed with finding out when alcohol sales start in state A, which is Maine.
<tool_call>
{"name": "search", "arguments": {"query_list": ["when do alcohol sales start in state Maine"]}}
</tool_call>
<tool_response>
{"result": "Search query:when do alcohol sales start in state Maine.
Doc 1 (Title:  "Alcohol laws of Maine") week.  On New Year's Day, alcohol may be sold until 2 a.m.  It may
be consumed in establishments licensed for on-premises consumption until 1:15 a.m., with the exception of New
Year's Day when it is 2:15 a.m.  Alcohol sales were previously prohibited until 9 a.m.  on Sundays.  In 2013
an exception was made to allow sales starting at 6 a.m.  when St.  Patrick's Day falls on a Sunday.  But in
2015 the legislature revised the law again to allow sales on any Sunday starting at 5 a.m., the same as every
other day of the week.  It is illegal to.
Doc 2 (Title:  "Alcohol laws of Maine") popular spirits in an attempt to prevent the loss of sales to New
Hampshire, which has lower prices and no sales tax.  The Maine Department of Public Safety, through the Maine
State Police's Liquor and Licensing Division, is responsible for licensing the manufacture, importation,
storage, transportation and sale of all liquor.  They also administer those laws relating to licensing and
the collection of taxes on malt liquor and wine.  Title 28-A of the Maine Revised Statutes contains the laws
and rules regulating liquor.  Alcohol may be sold between the hours of 5 a.m.  and 1 a.m.  each day of the.
Doc 3 (Title:  "Alcohol laws of Maine") it is prohibited to use an image of the Maine State House in liquor
advertising.  The legal age to purchase and consume alcohol in Maine is 21, which was instituted in 1987.
Minors may consume alcohol at home in the presence of the minor's parents.  Minors may not transport liquor
unless required due to the minor's employment, or at the request of their parent, guardian, or custodian.
Taxes on liquor at store level are included within the price and calculated by BABLO in order to raise enough
money to cover their expenses and generate money for the State General Fund"}
</tool_response>
Doc 1 states that alcohol sales in Maine start at 5 a.m.  on weekdays and on Sundays.  Now I will call answer
expert llm to generate the final answer based on the conversation context.
<tool_call>
{"name": "call_answer_llm", "arguments": {}}
</tool_call>
<Response>: 5 a.m.
```

864
865
866
867
868
869
870
871
872
873
874
875
876
877
878
879
880
881
882
883
884
885
886
887
888
889
890
891
892
893
894
895
896
897
898
899
900
901
902
903
904
905
906
907
908
909
910
911
912
913
914
915
916
917

## B PROMPTS USED IN EXPERIMENT

**Prompt for Process Reward**

You are a professional evaluation expert for multi-round search planning in RAG systems. Please provide a comprehensive score for the entire search planning process.

## Input Format
You will receive a complete message conversation flow, including:
- User's previous conversation content (optional)
- User question
- Assistant's multi-round search planning (reflected through tool_calls)
- Search results from each round (reflected through tool returns)
- Final call to call_answer_llm to end the planning process

## Scoring Dimensions and Deduction Standards
### [Reference Resolution Quality] Key Assessment
- **Severe issues (deduct 3 points)**: Any round of search queries contains unclear references
  - ✗ "its price" "how to apply for this policy" "the conditions mentioned above"
  - ✗ Search queries in multi-round conversations still contain reference words like "just mentioned", "previous", "this"
  - ✗ Second round searching "what are its competitors" without clarifying what "it" refers to
- **Medium issues (deduct 2 points)**: Inconsistent reference resolution across rounds
- **Good performance (no deduction)**: All rounds of search queries can be independently understood and executed
### [Multi-round Search Strategy Coherence] Key Assessment
- **Severe issues (deduct 3 points)**: Multi-round search lacks logical coherence
  - ✗ First round search results are sufficient, but still repeatedly searching for the same information
  - ✗ Ignoring previous search results, subsequent searches have no correlation
  - ✗ Over 50% of queries are highly repetitive across different rounds
- **Medium issues (deduct 2 points)**:
  - 2-3 rounds of obvious repetitive searching
  - Search strategy is too jumpy, lacking progressive logic
- **Minor issues (deduct 1 point)**: Individual rounds slightly repetitive but overall strategy is reasonable
- **Good performance (no deduction)**:
  - ✓ Adjusting subsequent query strategies based on search results
  - ✓ Different rounds cover different angles or delve into different aspects
  - ✓ Reflects information progression or supplementary logic
### [Search Precision]
- **Too broad (deduct 1 points per round)**:
  - ✗ "Shenzhen residency" (should add specific conditions/time)
  - ✗ "phone recommendations" (should add budget/requirements)
- **Too specific (deduct 1 points per round)**:
  - ✗ "December 2024 Shenzhen Nanshan District residency policy article 3"
### [Information Utilization and Completion Efficiency]
- **Severe issues (deduct 2 points)**:
  - Found key information but did not effectively utilize it in subsequent rounds
  - Obviously need supplementary information but did not conduct targeted searches
- **Medium issues (deduct 1 point)**:
  - Insufficient information utilization, redundant searching exists
  - Completion strategy not precise enough
- **Good performance (no deduction)**:
  - ✓ Discovering information gaps based on search results and precisely supplementing
  - ✓ Effectively utilizing obtained information to guide subsequent searches
### [Coverage Completeness and Termination Timing]
- **Missing core requirements (deduct 2 points)**:
  - The entire planning process did not cover user's core requirements
  - Far from user's intent
- **Inappropriate termination timing (deduct 1 point)**:
  - Terminating prematurely when information is insufficient
  - Continuing ineffective searches when information is already sufficient
- **Single perspective (deduct 1 points)**:
  - Multi-round searches still only query from a single perspective
### [Timeliness Consideration]
- **Missing time constraints (deduct 0.5 points per round)**:
  - For time-sensitive queries, should combine current time and add time constraints
- **Unnecessary time constraints (deduct 0.5 points per round)**:
  - Mistakenly adding time constraints for non-time-sensitive requirements

## Scoring Levels
- **5 points**: Excellent performance in all dimensions, efficient and coherent search strategy, sufficient information utilization, no obvious defects
- **4 points**: Overall good, basically reasonable search strategy, good information utilization (total deduction $\leq$ 1 point)
- **3 points**: Basically usable, search has certain strategic nature but average efficiency (total deduction around 2 points)
- **2 points**: Obvious problems exist, insufficient strategic nature or low efficiency (total deduction around 3 points)
- **1 point**: Seriously non-compliant, lacking effective search strategy (total deduction $\geq$ 4 points)

## Output Format
#### [Score]
Strictly follow the deduction standards, focus on the coherence of multi-round search strategies, information utilization efficiency, and entity decomposition completeness. Do not output any explanations and analysis, only output the score according to the format.

**Prompt for Answer Accuracy**

You are a seasoned Q&A expert tasked with evaluating the quality of a response based on the user's question, the standard answer, and the provided response.

User Question: `<question>`
Standard Answer: `<ground_truth>`
Provided Response: `<solution_str>`

Criteria: Based on the standard answer, you will determine whether the meaning of the provided response aligns with the standard answer. If they are consistent, the response is yes; otherwise, it is no.

Please directly answer with "yes" or "no". Do not include any explanation.

## C  LLM USAGE DISCLOSURE

In this work, the LLM is used only for two tasks: (1) generating the final answer based on the search planning trajectory; (2) scoring the consistency between the model-generated answer and the ground-truth answer.

