# OpenReview forum: "AI-SearchPlanner: Modular Agentic Search via Pareto-Optimal Multi-Objective Reinforcement Learning"
_ICLR.cc/2026/Conference — ICLR 2026 Conference Withdrawn Submission_

### Official Review · Reviewer_iLng · 2025-10-17

**Soundness:** 2
**Presentation:** 2
**Contribution:** 1
**Rating:** 0
**Confidence:** 4

**Summary:**

This paper introduces AI-SearchPlanner, a new RL algorithm for training LLM-based search agents. The core of AI-SearchPlanner is built on two key components: (1) a decoupled architecture that separates the search planner from the generator, and (2) a reward function designed to balance search effectiveness with computational cost. AI-SearchPlanner achieves superior performance to direct inference, RAG, and recent training methods.

**Strengths:**

I cannot find any strengths.

**Weaknesses:**

This paper has a critical flaw in its novelty. Nearly all of the claimed contributions originate from s3. The idea of decoupling the search planner and generator was first introduced in s3, and the reward function designed here is merely a simple tweak of the "A Gain Beyond RAG" reward proposed in s3.

The only difference from s3 is how the search planner's output is processed before passing to the generator. However, the compression-based method used in s3 would appear more cost-efficient, which contradicts this paper's central claims of efficiency. For these claims to be considered valid, a quantitative analysis is required. The authors should report precise input and output token counts and calculate the resulting FLOPS.

I am aware that s3 was recently accepted to EMNLP this August, so a direct comparison was not strictly necessary. However, this does not justify adopting its core ideas without significant novel contributions. As it stands, the paper does not meet the bar for publication due to its limited contribution. I advise the authors to develop and integrate a more substantial and original methodology for consideration at a future venue.

**Questions:**

I have no questions.

---

### Official Review · Reviewer_Mfu2 · 2025-10-28

**Soundness:** 2
**Presentation:** 2
**Contribution:** 2
**Rating:** 4
**Confidence:** 4

**Summary:**

The paper addresses the limitations of current "end-to-end single LLM handling both search planning and answer generation" approaches by proposing a decoupled architecture: a small model serves as the Search Planner, while a frozen large model acts as the Generator. It further introduces a dual-reward mechanism (“Outcome-Process” rewards) and a Pareto scalarization objective (Utility-Cost) to optimize the planning strategy using PPO, reporting strong performance and transferability on Wiki and Web datasets.

**Strengths:**

1. Dual-reward alignment: By simultaneously employing an Outcome Reward and a Process Reward, the framework constrains both “whether real gains are achieved” and “whether the planning trajectory is reasonable.” This enhances the coordination of retrieval and reasoning for multi-hop and cross-domain questions.
2. Cost-aware optimization: Planning utility and cost are incorporated into a unified optimization objective (controlling iterations and subquery numbers), providing a clear utility-cost trade-off curve. The framework achieves higher accuracy under similar costs or effectively reduces overhead while maintaining precision.

**Weaknesses:**

1. The idea of decoupling search and QA is not novel. Prior work, such as s3[1], has already separated the search planner from the generator. Although the authors mention s3 in Related Work, the paper does not articulate a distinctive innovation beyond that line. Consequently, the intro’s claim of novelty is unconvincing, and “decoupling search from QA” should not be counted as a contribution

2. Missing reproducible comparison against the most similar decoupled approach (s3).

3. Insufficient baselines from recent RL-for-search methods. Beyond the current comparisons, the paper should include stronger or orthogonal baselines such as ZeroSearch[2] and DeepResearcher[3] to more fairly position the contribution.


Process reward is computed by the frozen generator via a scoring prompt, which raises two biases:

4. Using the same (or same-family) LLM as both answerer and judge can encourage “planner behaviors that cater to the judge” rather than true information gain.

5. LLM-as-Judge variance/bias can amplify noise into the policy gradient. The paper should report human–LLM agreement, cross-judge robustness (different LLM families and prompts), and failure cases illustrating “reward hacking.”


6. Evaluation focuses mainly on Wiki-QA and WebShaper/WebWalker-style data. It lacks tests on more open-ended, long-horizon, high-uncertainty benchmarks and comparisons to representative “deep research” systems. If the paper claims industrial applicability, robustness and efficiency should be validated in truly open environments.

7. More experimental evidence is needed. For instance, restricting the retriever to E5 with k=3 may bottleneck multi-hop performance—please study different retrievers and k. Also report end-to-end latency and API cost, as well as training steps, throughput, memory footprint, and wall-clock training time to substantiate the “efficiency/cost” claims.


[1] You Don’t Need That Much Data to Train a Search Agent via RL

[2] ZeroSearch: Incentivize the Search Capability of LLMs without Searching

[3] DeepResearcher: Scaling Deep Research via Reinforcement Learning in Real-world Environments

**Questions:**

Please address the issues mentioned in the Weaknesses.

---

### Official Review · Reviewer_T4K7 · 2025-10-30

**Soundness:** 2
**Presentation:** 2
**Contribution:** 2
**Rating:** 4
**Confidence:** 5

**Summary:**

This paper explores an interesting problem of training a search agent system with reinforcement learning. Different from existing mainstream RL method, which relies on a single LLM agent to conduct both searching and answering, the authors propose to disentangle the search component and answer component into two separate LLMs and only optimize the search component with RL. Experiments are conducted in several benchmark datasets to demonstrate the effectiveness of the proposed method.

**Strengths:**

1. The problem studied in this paper is real and important.
2. The author proposed to disentangle the search and answer components into two LLMs, which is reasonable.
3. Experiments are conducted to demonstrate the effectiveness of the proposed method.

**Weaknesses:**

1. The paper is not well-written, and many parts lack further explanation. For example, it is unclear what the Score() function is in Eq.(1) and how to get a score from Eq.(3). It is not mentioned what model is used as LLM_gen in Eq.(3);
2. I am not very convinced by the novelty of this work. It seems to me that the biggest novelty compared to existing works, such as Search-R1, is to disentangle the search LLM and the answering LLM. However, this is also introduced in the S3 paper.
3. I would question whether the comparison with the baseline methods is fair. In your method, you are relying on a bigger size LLM (32B+) as the generator, while in the baselines, such as Search-R1, Search-o1, the model size is quite small (7B). The comparison is not fair in this sense.

**Questions:**

1. What is Score() and how to get the score from Eq.(3)?
2. What is the distinguishing novelty of this paper?
3. Is the comparison with baselines fair?

---

### Official Review · Reviewer_9SPn · 2025-10-31

**Soundness:** 2
**Presentation:** 3
**Contribution:** 2
**Rating:** 4
**Confidence:** 4

**Summary:**

Motivated by the challenge to optimize a single model for search planning and QA generation, the paper develops AI-SearchPlanner, which decouples the two functions. They also proposes the dual reward alignment which combines the outcome and process rewards. The experiment suggests that decoupling approach achieves better performance comparing to the non-decoupled planner-generator approach.

**Strengths:**

The paper shows the advantage of decoupling the search planner and QA generation model comparing to non-decoupling approach. The writing is clear and easy to follow. The ablation study is thorough, which highlight the importance of process reward and strong transferability of the search planner model across different datasets.

**Weaknesses:**

The experiment baseline is too weak. Though it includes the naive RAG with large model (deepseek and Qwen3-32b) as the baseline (table 1), it does not compare to using these models for search planning + QA generation. Even without further finetuning, these models also have the capacity to perform search planning. Moreover, since the paper emphasized the trade-off between efficiency and effectiveness, deploying a single model for search QA has significant advantage in system level comparing to deploying two different models. The comparison on efficiency needs careful evaluation.

**Questions:**

It would be helpful to add a stronger baseline (see weakness part) and perform more complete evaluation of efficiency-effectiveness trade-off.

---

### Note · Authors · 2025-12-08

I have read and agree with the venue's withdrawal policy on behalf of myself and my co-authors.